# Green Technology for Remediation of Water Polluted with Petroleum Crude Oil: Using of *Eichhornia crassipes* (Mart.) Solms Combined with Magnetic Nanoparticles Capped with Myrrh Resources of Saudi Arabia

**DOI:** 10.3390/nano10020262

**Published:** 2020-02-04

**Authors:** Ayman M. Atta, Nermen H. Mohamed, Ahmad K. Hegazy, Yasser M. Moustafa, Rodina R. Mohamed, Gehan Safwat, Ayman A. Diab

**Affiliations:** 1Chemistry Department, College of Science, King Saud University, Riyadh 11451, Saudi Arabia; 2Egyptian Petroleum Research Institute, Nasr City, Cairo 11435, Egypt; nermenhefiny@yahoo.com (N.H.M.); ymoustafa12@yahoo.com (Y.M.M.); 3Botany and Microbiology Department, Faculty of Science, Cairo University, Giza 11435, Egypt; 4Faculty of Biotechnology, Modern Science and Arts University, 6th October City 11435, Egypt; rodina.refat@msa.edu.eg (R.R.M.); gehan.safwat@hotmail.co.uk (G.S.); aymanalidiab@gmail.com (A.A.D.)

**Keywords:** magnetite, nanoparticles, myrrh, oil spill, remediation

## Abstract

Crude oil pollution of water bodies is a worldwide problem that affects water ecosystems and is detrimental to human health and the diversity of living organisms. The objective of this study was to assess the ability of water hyacinth (*Eichhornia crassipes* (Mart.) Solms) combined with the presence of magnetic nanoparticles capped with natural products based on Myrrh to treat fresh water contaminated by crude petroleum oil. Magnetic nanoparticles based on magnetite capped with Myrrh extracts were prepared, characterized, and used to adsorb heavy components of the crude oil. The hydrophobic hexane and ether Myrrh extracts were isolated and used as capping for magnetite nanoparticles. The chemical structures, morphologies, particle sizes, and magnetic characteristics of the magnetic nanoparticles were investigated. The adsorption efficiencies of the magnetic nanoparticles show a greater efficiency to adsorb more than 95% of the heavy crude oil components. Offsets of Water hyacinth were raised in bowls containing Nile River fresh water under open greenhouse conditions, and subjected to varying crude oil contamination treatments of 0.5, 1, 2, 3, and 5 mL/L for one month. Plants were harvested and separated into shoots and roots, oven dried at 65 °C, and grounded into powder for further analysis of sulphur and total aromatic and saturated hydrocarbons, as well as individual aromatic constituents. The pigments of chlorophylls and carotenoids were measured spectrophotometrically in fresh plant leaves. The results indicated that the bioaccumulation of sulphur in plant tissues increased with the increased level of oil contamination. Water analysis showed significant reduction in polyaromatic hydrocarbons. The increase of crude oil contamination resulted in a decrease of chlorophylls and carotenoid content of the plant tissues. The results indicate that the water hyacinth can be used for remediation of water slightly polluted by crude petroleum oil. The presence of magnetite nanoparticles capped with Myrrh resources improved the remediation of water highly polluted by petroleum crude oil.

## 1. Introduction

The leakage or spills of petroleum crude oil into water bodies occurs during the production, transportation, and consumption stages, resulting in adverse effects on the abiotic and biotic components of the ecosystem. According to the World Health Organization (WHO), almost 1 billion people in 60 countries lack access to uncontaminated water supplies. From 1970 to 2017, approximately 5.74 million tons of oil were released into water bodies around the world including rivers, lakes, and oceans [1]. Contamination of these water resources by petroleum hydrocarbons leads to significant consequences regarding human health and biotic components of the ecosystem; thus, it is considered a global problem that requires intervention. The contamination of surface and groundwater is a worldwide problem that affects biodiversity and undermines economic growth and the health of billions of people [2]. The impact of oil on marine environments is influenced by several factors, including the amount of oil, its chemical composition, the abundance of toxic compounds, and the proximity to environmentally, economically sensitive areas [1]. The characteristics which are incremental to determining the impacts of oil pollution include water depths, amounts of sediment, wind and tide action, degree of salinity, and water temperature. There are several ways in which living organisms can be exposed to oil pollution in aquatic environments, including direct dermal contact, absorption, ingestion, or inhalation of toxic components, as well as uptake through means of the food chain [2]. The toxicity of crude oil can be divided into acute or chronic effects. Acute toxicity is defined as short-term effects resulting from sudden exposure to high concentrations of oil. Chronic toxicity refers to effects resulting from long-term and continuous exposure to oil. Chronic exposure to crude oil results in alterations in the biological organization of an ecosystem and affects living organisms at the population and community levels. The concentration of oil released, duration of exposure, the tenacity and availability of specific hydrocarbons, and the extent to which other components of oil bioaccumulation in the environment are all different parameters that determine whether the effects will be acute or chronic. There are several strategies such as physical, mechanical, and chemical treatments which are used to alleviate pollution of the crude oil contaminants; however, the aforementioned treatments present consequences that are harsh on the environment and occasionally lack efficacy in action. Physical or mechanical removal provides only partial removal, whereas chemical-based methods result in additional toxic effects on the environment. Environmental research developments, particularly green technology, have proven to be an environmentally safe alternative for the remediation of polluted areas. Recently, magnetic nanomaterials based on magnetite have been attracting great attention due to their application as an oil-spill collector, with characteristics such as strong magnetic properties, low toxicity, and excellent biocompatibility [3,4,5,6,7]. Magnetic nanomaterials based on magnetite should be capped with hydrophobic polymers or surfactants to easily disperse in crude oil contaminants without dispersing in water. Moreover, magnetic materials should have superparamagnetic characteristics to effectively respond to an external magnetic field, and lose their magnetism after removal of that field [3,4,5,6,7,8,9,10]. Magnetic nanomaterials can be added to the crude oil spills either as powders or as a dispersed solution in heptane solvent, also known as magnetic or ferro-fluids [3,4,5,6,7,8,9,10]. It was previously reported that natural materials based on hydrophobic fatty acids, Arabic gum, rosin acids, and hydrophobic plants extracts when supplemented, stabilized the magnetite with a lower reduction of magnetic characteristics [8,9,10]. The remediation technologies that are currently available for removing oil from waterbodies are either combined with physical, chemical, bioremediation, or phytoremediation-based methods.

A combination of bioremediation with two or more methods to control oil spill pollution are often optimal for satisfactory removal of crude oil without environmental consequences. Bioremediation and phytoremediation are clean technologies, which bear minimal risk on the environment. Phytoremediation (also known plant-assisted bioremediation) is based on using plants and their associated rhizosphere to reduce the volume, mobility, or toxicity of contaminants in various environmental compartments including soil, groundwater, and other contaminated mediums [11,12]. This is considered to be a cost-effective, ideal method for the treatment of contaminated sites. Furthermore, phytoremediation is a clean technology which bears no risk of adverse environmental effects. Phytoremediation has been successfully applied to sites contaminated with sewage effluents, heavy metals, synthetic dyes, pesticides, crude oil, and polyaromatic hydrocarbons (PAHs) derived from various sources [12]. Water hyacinth (*Eichhornia crassipes*) is a submerged aquatic plant belonging to the family *Pontederiacea*. It is naturally native to tropical and subtropical South America, and particularly inhabits the Amazonian basin. However, water hyacinth can be found as an invasive species in various regions in the world including Europe, Africa, and Asia [13]. The plants themselves exhibit a vigorous growth rate, large biomass, and tolerance for many metal and metalloid elements, as well as organic compounds, making them an ideal option for phytoremediation. The utility of water hyacinth in the removal of several types of hydrocarbons from different media has also been investigated. Water hyacinth was used to degrade polycyclic aromatic hydrocarbons from soil leachate, achieving acceptable results [14]. It was also demonstrated that water hyacinth was able to accumulate hydrocarbons from contaminated sediments [14]. The present work aimed to use both water hyacinth and the constituents of Arabian Myrrh as an oleo-gum-resin obtained from the stem of different species of *Commiphora* as naturally occurring materials, in order to alleviate environmental pollution caused by oil spills. The present work demonstrated the application of water hyacinth as a more efficient method for the remediation of crude oil contaminants using oil spill collectors to remove the heavy components through the application of an external magnet. In this respect, magnetic nanomaterials based on magnetite were used as enhancers to accelerate the removal of petroleum oil spill pollutants from contaminated wastewater. Myrrh gum was used as an alternative biopolymer for the capping and stabilization magnetic iron oxide nanoparticles (MNPs) by physical adsorption and the establishment of chemical adsorption at the particle surface. The objective of the work is further achieved through modifying the innovative nanoparticles as oil spill collectors after applying an external magnetic field to accelerate the bioremediation in the presence of water hyacinth *(Eichhornia Crassipes)* (Mart. Solms). 

## 2. Experimental Methods

### 2.1. Materials

All chemicals used in the present work were purchased from Sigma-Aldrich Chemical Co. (Missouri, MO, USA) without further purification. Magnetite nanoparticles (MNPs) were prepared by reacting anhydrous ferric chloride, potassium iodide, and ammonium hydroxide (25%) as reported previously [8,9,10]. Myrrh gum is a local Saudi commercial natural product material (Riyadh, Kingdom Saudi Arabia) extracted from a tree yellowish red in color and covered with brownish yellow dust. It has a balsamic odor and a bitter, acrid taste. Myrrh was extracted sequentially with hexane and then ether three times for 72 h each at 25 °C, and then used as hydrophobic Myrrh extract after drying and removal of solvent by rotary evaporator. The water soluble fraction of Myrrh was extracted from ethanol/water (1:1 volume%) and used as capping agent after recovery using a vacuum evaporator. *Eichhornia Crassipes* plants produced in an open greenhouse (belonging to Cairo university, Egypt) and it spanned duration of four weeks with uniform height and rosette diameters. The average temperature in the greenhouse was between 18–26 °C and relative humidity was 20%–30%. The water used to maintain the plants was collected from different locations along the Nile River in order to provide nutrients and conditions found in nature. Egyptian heavy crude oil (API 19 degree) was produced from the eastern desert (Belayium petroleum company, Gulf of Suez, Egypt). Arabic medium crude was produced by ARAMCO (ARAMCO; Dammam, Kingdom of Saudi Arabia). Their specifications were summarized and presented in Table 1. The seawater with the total dissolved salt (36,170 mg·L^−1^) was collected from the Arabian Gulf along the Saudi coast.

### 2.2. Capping of Magnetite in the Presence of Myrrh 

The capping of MNPs with Myrrh water soluble or hydrophobic extract was prepared as follows: anhydrous FeCl_3_ (40 g) solubilized in water (300 mL) and added to potassium iodide solution (13.2 g dissolved in 50 mL of distilled water) at room temperature under stirring for one hour and N_2_ atmosphere. The filtrate was separated after removal of iodine precipitate and heated at 45 °C. The hydrophilic or hydrophobic Myrrh extract was (10 g solubilized in 100 mL of ethanol) added dropwise at the same time with 25% ammonia solution (200 mL). The reaction was continued at the reaction temperature and stirred for 4 h. The MNPs capped either with hydrophilic or hydrophobic Myrrh extracts were collected using an external magnet (a permanent Nd-Fe-B magnet 4300 Gauss). The MNPs were washed with ethanol and air dried to get the reaction yield percentage of 99.5%.

### 2.3. Characterization 

The chemical structure of iron oxides was confirmed by Fourier infrared (FTIR Nicolet FTIR spectrophotometer; model Nexus 6700 FTIR, Thermo scientific, MN, USA) using KBr. The morphologies of iron oxides capped with Myrrh extracts were confirmed by transmission electron microscopy (TEM; JEOL JEM-2100F; JEOL, Tokyo, Japan) and scanning electron microscope (SEM; JEOL JXA-840A, JEOL, Tokyo, Japan). The particle size diameter, polydispersity index (PDI), and surface charges (zeta potential; mV) of iron oxides aqueous solution were analyzed by dynamic light scattering (DLS; Zetasizer 3000HS; Malvern Instruments, Malvern, UK). Magnetic characteristics of iron oxides were determined by using vibrating sample magnetometer (VSM; LDJ9600 in a field of 20,000 Oe, LDJ Electronics, MI, USA). 

Field ionization mass spectrometry (FIMS) is a recommended technique to analyze complex multicomponent hydrocarbon mixtures (n-paraffins, iso-paraffins and cycloparaffins) without using a prior separation technique [15]. In this respect, a capillary gas chromatography-field ionization mass spectrometer, CCC-FIMS, was used to determine the composition of the remaining crude oil residual carried out by means of an HP 589041 series CC connected to a mass selective detector (MSD) 5970 (Hewlett-Packard, Palo Alto, CA). It was allowed on ion monitoring mode (SIM) and A quartz capillary column (25 m × 0.32 mm × 0.52 micro-m film Ultra-2) used for analysis. The system control and data acquisition were executed by an HP 59970 ChemStation (Hewlett-Packard, Palo Alto, CA, USA). The operating conditions for CGC were as follows: helium carrier gas (1.4 mL·min^−1^); injector and detector temperature was 300 °C. The temperature was programmed as 0 min at 90 °C, then ramped to 270 °C at 6 °C·min^−1^, and held for 30 min at 270 °C. The injected quantity was 1 µL of 2% volume solution in tetrachloromethane of the spill sample. 

### 2.4. Bioremediation of the Crude Oil Using Eichhornia Crassipes Plants 

Five different treatments were prepared in this experiment: Treatment A- (Plant + 0.5 mL crude oil), Treatment B- (Plant + 1 mL crude oil), Treatment C- (Plant + 2 mL crude oil), Treatment D- (Plant + 3 mL crude oil), and Treatment E- (Plant + 5 mL crude oil). *Eichhornia crassipes* plants added to Nile water, without addition of any crude oil, were used as a positive control. Each treatment was tested on three replicates. 

Chlorophyll and carotenoid extraction were conducted for samples, with 3 replicates for each crude oil concentration, following the protocol described as [16]: Three leaf disks (3 cm^2^; approx. 0.2181 g dry weigh) were randomly sampled from each leaf and added to test tubes containing dimethyl sulfoxide (DMSO). The samples were stored at 4 °C until complete extraction of the pigments. The residual plant material was filtered out using glass funnels and DMSO was used to standardize the sample volume at 15 mL for each sample. The absorbance was measured using UV-visible spectrophotometer (Shimadzu UV-1208 model; Canby, OR, USA) at the following wavelengths: 436, 440, 474, 644, and 662 nm, calibrating to zero with pure DMSO. The chlorophyll and carotenoid contents (μg.g ^−1^ dry weight) were determined using the Equations (1)–(4):(1)$$\mathrm{Chl}\;a\;=\;0.0127\;A_{266}\;+\;0.02269\;A_{644}(g\;L^{-1})$$
(2)$$\mathrm{Chl}\;b\;=\;0.0229\;A_{622}\;+\;0.00468\;A_{644}(g\;L^{-1})$$
(3)$$\mathrm{Total}\;\mathrm{chlorophyll}\;=\;0.0202\;A_{622}\;+\;0.00802\;A_{644}(g\;L^{-1})$$
(4)$$C_{x+c}\;=\;(1000A_{474}\;-\;1.29\;\mathrm{Chl}\;a\;-\;53.77\;\mathrm{Chl}\;b)/220(g\;L^{-1})$$
where Chl a = chlorophyll a; Chl b = chlorophyll b; Chl a + Chl b = total chlorophyll; $C_{x+c}$ = carotenoids; $A_{x}\;$= absorbance at *x* nm.

After four weeks, the plants were harvested, thoroughly washed, and dried in absorbent paper. Each plant was divided into emerged parts (stem and leaves) and submerged parts (roots). Each sample was dried at 75 °C using a dry oven and grinded using an electric mill. The replicates of each sample were mixed together for both the emerged and submerged parts. Sulphur concentration in the separated shoots and roots of *Eichhornia crassipes* were measured using an X-ray fluorescence sulphur meter according to the method described by the American Society for Testing and Materials (ASTM D-4294-98).

### 2.5. Application of MNPs as Oil Spill Collectors

In a 500 mL beaker, 1 mL of heavy crude oil was poured over 250 mL of seawater. Different weight ratios of MNPs to crude oil, ranging from 1:1 to 1:50, were added and mixed slowly with the crude oil over the seawater for 1 min using a glass rod. After 5 min, a permanent Nd-Fe-B magnet (4300 Gauss) was used to collect the dispersed crude oil spill. The remaining oil was extracted from the seawater by using chloroform. Liquid-liquid extraction technique was utilized to extract crude oil from water. Chloroform was used as organic solvent in the extraction process. Chloroform and water were added to a separating funnel and the oil was allowed to separate from water. The high-density fraction containing the oil and the solvent was then recovered with aluminum foil and stored until complete evaporation of the solvent. The oil spill collection efficiency CE% of the MNPs was calculated using the following Equation (5): CE% = V0/V1 × 100(5)
where V0 and V1 are the volume of the removed and original oil, respectively. The used MNPs were recycled after collection with an external magnetic field and washing them several times with chloroform.

The aromatic and saturated components of the crude petrolatum were determined using liquid–solid column chromatography according to the method described by the American Society for Testing and Materials (ASTM-D271). In this respect, a column with a 1.3 cm diameter and a height of 130 cm was packed with activated silica gel (60–200 mesh size). The crude oil (10 g) dissolved in a few milliliters of heptane was transferred to the column and eluted with 300 mL of heptane, followed by 200 mL of benzene, and finally 100 mL of a 1:1 mixture of absolute methanol and benzene. The solvents were distilled and the refractive index of each fraction was determined. The elutes were combined into saturated, mono-, di-, and poly-aromatics according to the refractive index data at 20 °C. The saturated hydrocarbons, mono-cyclic, bi-cyclic, and poly-cyclic aromatics have refractive indices below 1.48, from 1.48 to 1.52, 1.53 to 1.58, and higher than 1.59, respectively. All data were expressed as means of three replicates ± standard error (SE). Statistical analysis was performed using one-way Analysis of Variance (ANOVA) and Least Significant Difference (LSD) test to determine differences between the group’s mean and standard error at 0.05 levels. Frequencies were computed and Chi-square test was utilized to study the association between the different variables. All statistics were carried out using the Statistical Analysis Systems (SAS) program. Differences were considered significant at *p* < 0.05. The least significant difference (LSD) was determined according to the following Equation (6): (6)$$LSD=t_{critical}\left(0.0512\right)\sqrt{Msw(\frac{1}{n}}+\frac{1}{n})$$

The results include content analysis for photosynthetic pigments in the leaf blades of *Eichhornia Crassipes* plants, were measured at different levels of contamination, and will be accompanied by a morphological assessment of plants (refer to index 1 for full data on results of pigment analysis). The results also include comparative analysis of sulphur concentrations in plant shoots and roots. Finally, the saturated and polyaromatic hydrocarbons, as well as individual aromatic constituents in remediated water at the end of the study, will also be demonstrated. 

## 3. Results 

The iron oxides were previously prepared by using co-precipitation based on FeCl_3_ and KI, either with removal of iodine or without, to produce several types of iron oxides after hydrolyzing at a temperature of 45 °C and adjusting the pH to approximately 8.5–9 [8,9,10]. In this respect, iron oxides were prepared in the absence of a capping agent or in the presence of Myrrh extracts. Trees of certain *Commiphora* species of the *Burseraceae* family produced resinous exudates known as Myrrh. It is well established that the Myrrh contains a 2%–8% volatile oil (myrrhol), 23%–40% resin (myrrhin), 40%–60% gum, and a bitter principle 10%–25% [17,18]. The composition of the volatile oil based on steroids, sterols, and terpenes such as furanogermacrens and the two furanoeudesmanes (minestrone and furanoeudesma1,3-diene) besides fatty acids that extracted as hydrophobic extracts by using heptane [17]. The hydrophobic Myrrh resins extract based on ether fractions consist of α-, β- and γ-commiphoric acid, esters of a resin acid, commiphorinic acid, and two phenolic resins, α- and β-heerabomyrrholic acid [17]. The hydrophilic extracts consist of water-soluble gum, alcohol-soluble resins, and volatile oil beside polysaccharides, amino acids, and proteins [17]. The addition of capping agents modified with carboxylate and hydroxy carboxylate ions or polysacharides during the formation of magnetite assists the formation of nanoparticles with controlled sizes and morphologies. Myrrh is used as a capping agent without separation of its components or with extraction of hydrophilic or hydrophobic extracts as represented in the experimental section. The adsorption of Myrrh on iron oxides in the basic medium can occur due to physical and chemical adsorptions of the hydroxyl groups located on the iron oxide with Myrrh carboxylic, amine, hydroxyl, and aldehyde groups [18]. 

FTIR spectra of iron oxides without capping and with Myrrh, hydrophobic extract, and hydrophilic extract were represented in Figure 1a–d, respectively. All FTIR spectra confirm the appearance of one peak at around 570 cm^−1^ assigned for Fe-O stretching of Fe_3_O_4_ without appearance of bands at 650 and 750 cm^−1^ (Fe–O maghemite) to elucidate stability of magnetite for further oxidation [19]. The disappearance of a strong band at 3450 cm^−1^ (assigned to the structural OH stretching) of iron oxides capped with hydrophobic extract (Figure 1c) confirms the hydrophobicity of MNPs. The appearance of a broad band at 3450, 1630, and 1550 cm^−1^ which is attributed to OH, COOH, and C=C stretching of hydrophilic Myrrh extracts (Figure 1b,d) reveals the hydrophilic functionalization of magnetite nanoparticles with carboxylic or hydroxyl groups of Myrrh. Consequently, the prepared MNPs in the presence of hydrophobic extracts were stabilized via electrostatic attraction between the carboxylic groups of hydrophobic fatty acids of Myrrh gum or the glycoprotein and the hydroxyl group of Fe_3_O_4_ [19]. 

The particle sizes and polydispersity index (PDI) of MNPs prepared in the absence and presence of Myrrh extract as a capping agent were determined in heptane by dynamic light scattering (DLS) and represented in Figure 2a–d. The diameter of the iron oxide nanoparticles in the absence of Myrrh (Figure 2a) agglomerated in heptane and were found to be 67.3 nm and PDI at 0.438 to confirm their hydrophilicity which referred to the presence of the associated and hydrated layer of magnetite [20]. The capping of iron oxides with Myrrh reduced PDI and diameters to be 0.283 and 8.3 nm (Figure 2b), correspondingly in heptane to interpret the hydrophobicity of iron oxides NPs. The capping of MNPs with hydrophilic extract increased the particle sizes to 177.4 nm (Figure 2c) in heptane and confirms the formation of hydrophilic capping agents on MNPs. The hydrophobicity of MNPs in the presence of the hydrophobic Myrrh extracts reduced their PDI and diameters to 0.128 and 39.6 nm (Figure 2d), respectively, to elucidate the mono-dispersity and hydrophobicity of MNPs in heptane after capping with hydrophobic Myrrh extracts.

The morphologies and surface morphologies of MNPs prepared in the absence and presence of Myrrh extracts were evaluated by using TEM and SEM micrographs as summarized in Figure 3a–d and Figure 4a–d, respectively. The cubic, rods, stretched spherical, and spherical morphologies were observed for MNPs prepared in absence of capping, Myrrh, hydrophilic and hydrophobic Myrrh extracts as appeared in Figure 3a–d and Figure 4a–d, respectively. The rods appeared in Figure 4b appeared as spherical particles agglomerated as rods (Figure 3b) to elucidate that the presence of both hydrophilic and hydrophobic extracts of Myrrh as capping agents agglomerated the MNPs in rod morphology. The particle sizes distribution appeared in Figure 3a–d and Figure 4a–d agree with the data reported in DLS measurements (Figure 2a–d). Accordingly, the variations in the compositions of hydrophobic and hydrophilic Myrrh extracts controls both the morphologies and particle sizes of MNPs. This behavior was referred to interaction of hydrophilic moieties of Myrrh extract that contains carboxylic, hydroxyl, aldehyde and amine functional groups with the hydroxyl groups of MNPs control both their morphologies, particle sizes distribution, and diameters (Figure 2a–d, Figure 3a–d and Figure 4a–d). The particle size distributions appear to be quite narrow and the particle shape is reasonably spherical in the presence of hydrophobic Myrrh extract to elucidate the formation of monolayer of Myrrh capping agents on the MNPs surfaces [3].

The magnetic characteristics of the synthesized MNPs synthesized in the presence and absence of the Myrrh extract such as the saturation magnetization (Ms), magnetic remanence (Mr), and coercivity (Hc), were measured at room temperature by VSM magnetic hysteresis loops. The results are shown in Figure 5 and in Table 2. The higher Ms and lower Mr and Hc of all MNPs (Table 1 and Figure 5a–d) elucidate their superparamagnetic properties of MNPs capped either with Myrrh and their hydrophobic Myrrh more than that capped with the hydrophilic extracts.

The CE% of MNPs capped with Myrrh and its hydrophobic extract in the collection of an oil spill clean-up of Arabian medium crude oil was evaluated at different MNPs to crude oil ratios (1:1 to 1:50) and are summarized in Table 3. The oil spill was reused five times, as described in the experimental section, without change in the CE%. The chemical compositions of the remained crude oil residuals after using MNPs: hydrophobic Myrrh extract (1:10) were separated by extraction with chloroform and analyzed by using CGC-MS according to the ASTM D 4489-85 and Nord test Method NT CHEM 0011. The CGC-MS data of saturates of medium crude were summarized in Table 4. The data summarized in Table 4 show the distribution of n-alkanes, isoprenoids, polycyclic aromatic hydrocarbons and their alkyl-homologues, and biomarker compounds (hopanes and steranes) using a mass range of m/z 113-200.

The analysis of *E. crassipes* plants subjected to different treatments of crude oil is represented below in (Figure 6). The hydrocarbons contents included paraffin, saturates, aromatics and total hydrocarbons in Nile water were measured after plant treatments. While the sulphur contents in the roots and leaves were measured inside the plant roots and leaves. Plants subjected to 1 mL, 2 mL, 3 mL and 5 mL of crude oil all showed significant reduction in carotenoid content compared to the control. Treatment B had the lowest concentration of carotenoid pigments in the plant leaves. While, subsequent treatments registered higher concentrations of carotenoids but failed to exceed control levels. The plant seemed to be tolerant to 0.5 mL crude oil since this treatment failed to cause any significant reductions in total carotenoid concentration. 

The concentrations of chlorophyll a (Chl a) and chlorophyll b (Chl b) were measured in *Eichhornia crassipes* plants under different crude oil treatments and represented in Figure 7. The concentration of chlorophyll a and chlorophyll b in the control treatment (plant without addition of crude oil) were 1728.3 ± 95.27 μg/g dry weight and 641.27 ± 35.03 μg/g dry weight, respectively. Plants subjected to 0.5 mL, 1 mL, 2 mL, 3 mL and 5 mL oil showed significant reduction in the concentrations of both pigments. The lowest concentration of chlorophyll a and chlorophyll b were recorded in plants subjected to 5 mL crude oil. However, chlorophyll a seemed to be more affected by crude oil than chlorophyll b. In addition, chlorophyll b seemed to show moderate levels of adaption as the concentration of oil increased. The ratio of chlorophyll a to chlorophyll b in *E. crassipes* plants is represented in Figure 8. 

The curve (Figure 8) shows that the ratio gradually decreased beyond treatment A (plant + 1 mL oil), indicating that the concentration of chlorophyll a was more affected by the increasing level of crude oil contamination than chlorophyll b. Eichhornia *crassipes* plants exhibited loss of pigmentation, withering, visible reduction in biomass and signs of necrosis as concentration of crude oil gradually increased. Plants in treatment A and B (0.5 and 1 mL, respectively) seemed tolerant and exhibited only slight changes in morphology. However, plants subjected to higher concentrations 2 mL, 3 mL and 5 mL exhibited increasingly severe alterations in morphology Figure 9.

The amount of sulphur in the root and shoot of *Eichhornia crassipes* plants was analyzed and represented in Figure 10. The control treatment (plant without oil) showed trace levels of sulphur. The amount of sulphur increased gradually in all, following oil treatments up to 1787.3 ppm and 608.5 ppm in the root and shoot of *Eichhornia crassipes*, respectively. In all treatments, sulphur showed preferential accumulation in the plant roots.

The concentrations of saturated and aromatic hydrocarbons measured in remediated water after 4 weeks are shown below in Figure 11. Treatment A (plant + 0.5 mL oil) showed a slight increase in saturated hydrocarbons up to 42.3% and an increase in total aromatics in water to 57.7%, to elucidate the uptake of aromatic hydrocarbons. The final treatment (plant + 5 mL oil) recorded the highest reduction in aromatic hydrocarbons from 58.05% in the control to 49.75%, indicating the best uptake by the plant. 

The values for individual aromatic constituents are shown below in Table 5. The amount of polyaromatic hydrocarbons decreased from 44.86 at the start of experiment to 26.5. while the contents of monoaromatic and diaromatic hydrocarbons increased from 8.13 and 15.2 to 5.06 and 8.13, respectively. This indicates the degradation of polyaromatic hydrocarbons by *E. crassipes* into monoaromatic and diaromatic constituents as the experiment commenced. 

## 4. Discussion

The chemical structure of MNPs and the concentration of the hydroxyl groups on their surface elucidated from FTIR spectra (Figure 1a–d) reveals that the hydrophobic extracts of Myrrh are completely capped on the surface of MNPs. The presence of reactive carboxylic, hydroxyl, and aldehyde groups in the chemical structures of Myrrh facilities the interaction of Myrrh with the hydrated hydroxyl groups on the surface of magnetite either with the chemical or electrostatic interactions in the basic medium at pH ranged from 8 to 10 [3,4,5,6,7,8,9,10]. The disappearance of the magnetite hydroxyl group contents (Figure 1d) when hydrophobic extracts of Myrrh capped the magnetite explains the formation of a monolayer of hydrophobic Myrrh extract on the surface of magnetite. Moreover, the presence of the hydrophilic Myrrh extract during the formation of magnetite demonstrates the formation of bi or multi layers of the hydrophilic extract which were confirmed from the appearance of the hydrophilic function groups in the FTIR spectrum of the magnetite, in the presence of hydrophilic Myrrh extract (Figure 1d). The disappearance of Fe-O peaks of other iron oxides (750–950 cm^−1^) elucidates that the Myrrh extract protected the magnetite to oxidize further for another iron oxide such as maghemite and hematite [6]. The formation of a monolayer of Myrrh hydrophobic extract changed the polydispersity and particle sizes of MNPs to monodisperse hydrophobic magnetite as confirmed from DLS data (Figure 2a–d). It was suggested that the presence of carbonyl groups (ester, aldehyde and ketone) in the chemical structures of the capping agent improved their binding with magnetite nanoparticles due to their electron-withdrawing effect. This binding increases the water dispersibility of magnetite NPs [8]. Moreover, the primary alcohol groups of polysaccharides were partially oxidized to the resultant carbonyl groups that provide sufficient protection and stability to magnetite nanoparticles beside their high water-dispersibility [7]. The morphologies of MNPs (Figure 3 and Figure 4a–d) confirm the change of morphology from cubic, rod, and spherical by the changing the composition of Myrrh from hydrophilic to hydrophobic extracts to acquire the highest efficiency of Myrrh as a capping agent depends on its composition [3,4,5,6,7,8,9,10]. The variation of morphologies also affects the magnetic properties of the MNPS (Table 1 and Figure 3a–d) to produce superparamagnetic magnetite NPs. The strong magnetic properties of magnetite capped with hydrophobic extract compared to hydrophilic Myrrh extract (Table 2 and Figure 5a–d) show the formation of monolayer hydrophobic capping agents [3,4,5,6,7,8,9,10]. The presence of multilayers on the surface of magnetite in the presence of hydrophilic Myrrh extract reduces its Ms value and magnetization due to the formation of thick coats on the magnetite surfaces [3,4,5,6,7,8,9,10]. The hydrophobic MNPs were selected to clean the oil spill and their CE% (Table 3) shows that those with hydrophobic extract were used as effective NPs to remove the aromatic, polyaromatics, cyclo-paraffin, and iso-paraffin constituents of the Arabic heavy crude oil according to CGC-MS data. Moreover, the remained residual of the oil-spill analysis shows the higher efficiency of MNPs capped with hydrophobic Myrrh extract (CE%; Table 3) to remove the cyclo-paraffin and iso-paraffin from the crude oil constituent. It also confirms the good compatibility of MNPs with the heavy crude oil beside good magnetic properties even at MNPs: crude oil ratio (1:10) enhanced their applicability and reusability to remove oil spill. The reusability of the prepared MNPs for several cycles elucidate their high stability to salinity of seawater and crude oil acidity. The presence of the hydrophobic monolayer on the magnetite surfaces facilitated their dispersion in the hydrophobic crude oil contaminants without dispersion in seawater. Consequently, increasing the hydrophobic magnetite in the crude oil due to their high dispersion into the crude oil enhances its response to the external magnet and CE% value. While the formation of the multilayer of hydrophilic Myrrh extract on the magnetite surfaces reduces their magnetic characteristic and concentrations in the crude oil contaminants due to their dispersion into seawater and reduces its CE%. The practical application of the MNPs as oil spill collectors is preferred among the different mechanical, chemical, and bioremediation techniques used to control the oil-spill pollution due to simplicity, cost effectiveness, and low toxicity, as well as the reusability and recovery of crude oils. In this instance, it is reported that [21], the chemical dispersants were one of the most effective response methods in deeper water in terms of impacts on the total crude volume although the toxicity of chemicals, recovery of the crude oil, and toxicity of crude oil dispersion required more efforts. It is also reported that, the mechanical methods based on skimmers and oil sobers removed about 7–10% of the total oil and also required more efforts depended upon the type of shorelines to clean-up after removal it to shore by mechanical method [21]. The technologies used to prevent oil from reaching sensitive coastal habitats. The continued remediation efforts recommended also to control pollution of the remaining crude oil although it required long time period. Accordingly, new technology needs to be incorporated into the remediation process not only to determine the effective method to apply under various conditions, but also to feed into the process of discovering the operational end point. The present work succeeded to use magnetite and Myrrh extract which are environmentally friendly non-toxic materials [3,4,5,17,18] to collect and recover the crude oil with CE% more than 95% (Table 3). The remained unrecovered crude oil can be collected by *Eichhornia crassipes* plants as discussed in the forthcoming section.

Crude oil is well known for its phytotoxicity and capacity to alter the morphological, physiological, and biochemical properties of plants. It was previously reported that microcosm experiments performed in situ in North Inlet Estuary near Georgetown, South Carolina, using Texas crude oil confirmed a decrease in chlorophyll *a* in phytoplankton as crude oil concentration increased from 10 to 100 microliters per liter [22]. The results of this work (Figure 7 and Figure 8) demonstrate that chlorophyll a, chlorophyll b, and carotenoid contents decrease in *E. crassipes* response to oil contamination. Chlorosis and burn symptoms on plant leaves at higher concentrations of oil were also observed and appeared to concomitant with the reduction in chlorophyll and carotenoid pigments. Chlorophyll plays a fundamental role in plant physiology and its productivity. There are five types of chlorophyll pigments that can be found in photosynthetic organisms: Chlorophyll a, b, c, d, and f. Chlorophyll a and g are the most abundant types in *Eichhornia crassipes* plants and are the primary pigments used in photosynthesis processes. Chlorophyll levels are altered and responsed to various conditions including weather, amount of sunlight, and the presence of environmental contaminants [23]. In fact, the reduction in chlorophyll content in plants has been used as an indication of environmental contamination [24]. For the control treatment (no crude oil), chlorophyll a concentration was 1728.3 ± 95.27 μg/g DW. All treatments A, B, C, D, and E registered progressive reduction in chlorophyll a content as 1003.4 ± 50.17, 661.0 ± 64.30, 585.6 ± 57.73, 507.1 ± 26.88, and 396.8 ± 6.12 μg/g DW (*n* = 3), respectively, when compared to the control. The reduction in chlorophyll a concentration was also reported in various studies [21,22,23,24,25,26,27]. The effect of different concentrations of crude oil on chlorophyll content in *Simmondsia chinensis* plants was reported and showed a similar reduction in chlorophyll a content in all treatments [21,22,23]. A decrease was also reported in chlorophyll a content in response to petroleum-induced stress in *Vigna unguiculata* [25,26,27]. The effect of oil-induced stress on two common plant crops: *Raphanus sativus L* (raddish) and *Triticum aestivum.* L (common wheat) was reported and confirmed that chlorophyll a content significantly decreased in *T. aestivum. L* [27]. Meanwhile, the chlorophyll a content was increased in *R. sativus. L.* plants [27]. Chlorophyll b shared an analogous trend with chlorophyll a in that it also decreased in all treatments compared to its respective control (641.27 ± 35.03 μg/g DW). However, unlike chlorophyll a, it did not show linear regression. In Treatment A and Treatment B chlorophyll b contents were 366.3 ± 20.53 and 387.37 ± 43.19 μg/g DW, respectively. In treatments C and D chlorophyll b content increased up to 553.03 ± 48.83, and 539.23 ± 33.23 μg/g DW, respectively. However, these values decreased to 380.33 ± 15.03 in the treatment E (Nile water + 5 mL oil). Similar inconsistent findings for the effect of petroleum hydrocarbons on chlorophyll b content have been previously reported [28,29]. Progressive reduction in chlorophyll b content in *Abelmoschus esculentus* (okra) plants in treatments of spent-engine oil ranging from 50–200 mL was also reported [30]. Other work reported that the chlorophyll levels were substantially modified due to structural damage to the cell membranes and chloroplasts [31]. The reduction in chlorophyll content of *E. crassipes* could also be attributed to interference of the oil with the ability of the plant to absorb water mineral nutrients such as iron, magnesium, and boron that are essential for chlorophyll synthesis. Such interference coupled with the reduced photosynthetic capacity that accompanies the reduction of chlorophyll may explain the necrosis and burn symptoms observed in *E. Crassipes* leaves at higher crude oil concentrations. Carotenoids are essential pigments found in photosynthetic organisms including higher plants. They play an important role in photosynthesis by transferring absorbed light energy to chlorophylls. They also provide photooxidative protection against reactive oxygen species (ROS) inevitably generated by chlorophylls during photosynthesis, especially under environmental stress conditions [32]. In this work the oil contaminated treatments showed a significant reduction in carotenoid content (Figure 6) compared to the control (5.7 ± 0.33 μg/g DW) at *p* < 0. 01. Treatment A and B showed only slight reduction, and the lowest carotenoid content, respectively (Figure 6). Carotenoid content increased in treatment C up to 4.53 ± 0.21 μg/g DW and it was recorded as the maximum value in all treatments. For the remaining treatments: D and E, carotenoid contents decreased compared to treatment C but remained above the carotenoid level registered in treatment B. These findings agree with the previous work [33] that tested the effect of crude oil concentrations (ranging from 0.5–2%) on carotenoid content of *Scenedesmus obliquus* (a type of green algae) and reported a pattern similar to the present work. It was also reported that carotenoid levels progressively decrease in *Canavalia ensiformis* in response to crude oil [34]. These findings were, however, inconsistent with the increase in catenoid levels in *R. sativus L.* subjected to crude oil stress [30]. The reduction in carotenoid levels may be the result of the production of reactive oxygen species (ROS) due to oil-induced stress [35] or referred to the disruption of the plant’s ability to accumulate pigment-lipoprotein complexes such as photosystem I [36]. These data were also observed as illustrated in the surface morphologies of plant photos (Figure 9) before and after treatment with different crude oil contaminates. Total chlorophyll a, chlorophyll b, and carotenoid pigment concentrations were significantly affected by the gradual increase of crude oil and were accompanied by morphological alterations including visible yellowing and curling of leaves and severe burn-like symptoms observed at higher oil concentrations (Figure 9).

Absorption of the petroleum crude oil materials such as organic sulphur and other pollutants from the water medium depends mainly on plant roots, which absorb and accumulate the pollutants and subsequently translocate them into other plant organs. Microorganisms associated with plants play a role in oil degradation and positively affect the absorption and accumulation of sulphur in plant roots [37]. The phytoaccumulation of sulphur (Figure 10) is one of the primary forms of ionic stress that limit agricultural productivity [38]. Egyptian heavy crude oil (Table 1) has sulfur contents 2.8 Wt.% that are associated with PAHs. Therefore, the level of sulphur uptake is used as an indicator of the reduction of PAHs and hence the plant’s success in phytoremediation. In this work, the analysis of the sulphur content in plant roots and shoots (Figure 10) showed a high degree of sulphur extraction by *E. crassipes*. The increase of the sulphur content in the shoot and root was concomitant with the degradation of polycyclic aromatic hydrocarbons (PAHs; Figure 11 and Table 5). The concentrations of sulphur in the root ranged from 15 to 1787.3 ppm and were found to be significantly higher than the concentrations in the shoot, which ranged from 10 to 608.5 ppm. This observation indicates that the sulphur preferentially accumulates in the roots. The hyper-accumulation of sulphur by various plant species has been reported by using *Gypsophila struthium, Helianthemun squamatum, and Lepidium subulatum* [38]. The uptake of sulphur by *Bassia scoparia* indicated that the sulphur concentrations in the shoots and roots were increased in proportion with the degradation of hydrocarbons [37]. These findings coincide with the outcome of this work. However, it was also reported that the concentration of sulphur decreased in the shoot of *Viciafaba L*. with an increase of petroleum contamination, which contradicted with the present work [39]. It was also found that the accumulation of sulphur in plants growing on media contaminated with petroleum-derived substances (e.g., sulphur) depended largely on the species of the plant [40]. The values for total saturated and aromatic hydrocarbons (Figure 11 and Table 5) were measured in contaminated water at the end of the experiment. Compared to the corresponding baseline values, the percentage of aromatic hydrocarbons decreased while the saturated hydrocarbons increased in all treatments. These results indicate that E.crassipes degraded aromatic hydrocarbons. The phytodegradation of polyaromatic hydrocarbons was also reported by including *Vetiveriazizanioides, Bidenspilosa, Eleusineindica,* and *Zea mays* plants [41,42]. 

The previous results added new valuable data and green technology for the collection of a petroleum crude oil spill by using MNPs in deep seawater in the presence of an external magnetic field. The green technology of the present work was referred to the using and preparing of ecofriendly chemicals based on cheap natural resources and nontoxic materials based on magnetite, Myrrh, and *Eichhornia crassipes* (water hyacinth) to remove hazardous and toxic petroleum crude oil water pollutants. The capping of magnetite with the low cost hydrophobic extract of natural product based on Myrrh extract, which is not toxic, and ecofriendly facilitates the dispersion of magnetite in the crude oil without dispersion in seawater with improved superparamagnetic characteristics. The using of MNPs to collect petroleum crude oil has a great advantage over chemical dispersants because the MNPs reclaim and recover the petroleum crude oil and are reused several times with the high CE% besides their low toxicity. The MNPs can be practically applied using new inexpensive devices such as magnetic sponges, hydrophobic meshes, and magnetic devices controlled with an electric field [43,44,45]. This work confirms also that the phytodegradation of the petroleum crude oil spill in Nile river water can be carried out in the presence of *Eichhornia crassipes* (water hyacinth), which is able to treat PAHs and concomitant the organic sulphur in crude oil polluted water. It is recommended that the combination of both MNPs in the presence of water hyacinth is an effective green technology for remediation of water polluted with petroleum crude oil. The mobility of heavy and petroleum crude oils having higher specific gravity improved in the presence of nanomaterials and leads to an improvement in oil viscosity and recovery [46]. Accordingly, the remaining residue of the petroleum crude oil contaminates can be easily phytoremediated by using water hyacinth plants. 

## 5. Conclusions

This work proves that the higher efficiency of MNPs capped with hydrophobic Myrrh extract to collect an oil spill using an external magnet to remove the heavy crude oil polyromantic constituents cyclo-paraffin and iso-paraffin from crude oil. It also confirms the good compatibility of MNPs with heavy crude oil beside good magnetic properties even at MNPs: crude oil ratio (1:10) enhanced their applicability and reusability to remove an oil spill. Moreover, *Eichhornia crassipes* (water hyacinth) is able to treat PAHs and concomitant sulphur contamination in the crude oil polluted water at levels of up to 5 mL. However, total chlorophyll a, chlorophyll b, and carotenoid pigment concentrations were significantly affected by the gradual increase of crude oil and were accompanied by morphological alterations including visible yellowing and curling of leaves, and severe burn-like symptoms observed at higher oil concentrations. Phytotoxicity of the crude oil could potentially affect the reduction in biomass in *E.crassipes* plants and threaten its success in phytoremediation. It is therefore recommended that the combination of MNPs as an oil spill collector can control and reduce the effect of the crude oil phytotoxicity on *E.crassipes* plants. Further studies should be conducted to investigate the effect of crude oil on the biomass, morphological parameters, and biochemical properties of *E. crassipes*. 

## Figures and Tables

**Figure 1 nanomaterials-10-00262-f001:**
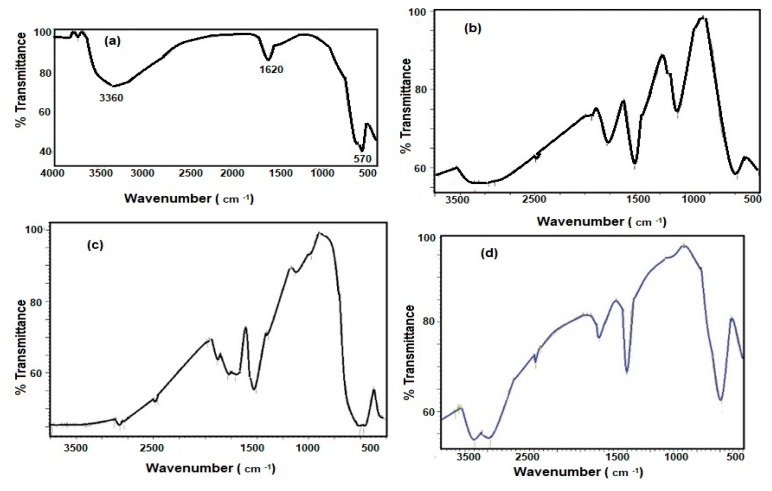
FTIR spectra of the prepared magnetite (**a**) absence of Myrrh, (**b**) presence of Myrrh, (**c**) presence of hydrophobic Myrrh extract and (**d**) presence of hydrophilic Myrrh extract.

**Figure 2 nanomaterials-10-00262-f002:**
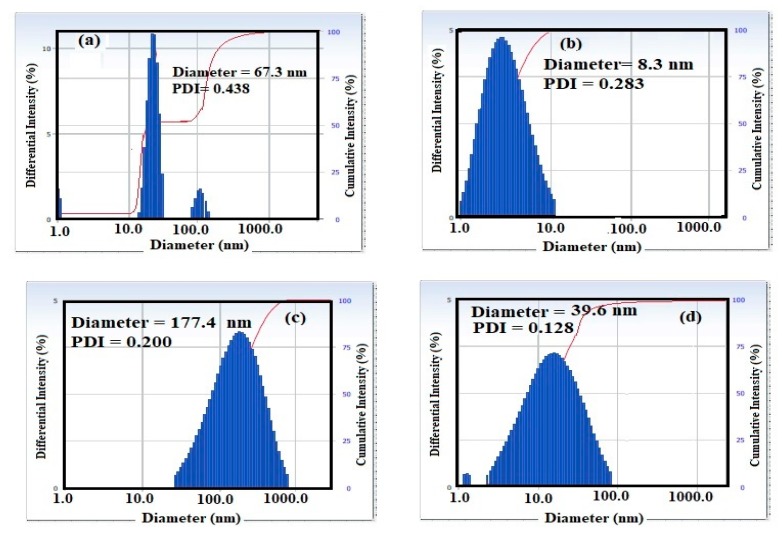
DLS of the prepared MNPs (**a**) absence of Myrrh, (**b**) presence of Myrrh, (**c**) presence of hydrophilic Myrrh and (**d**) presence of hydrophobic Myrrh extracts in heptane at 25 °C.

**Figure 3 nanomaterials-10-00262-f003:**
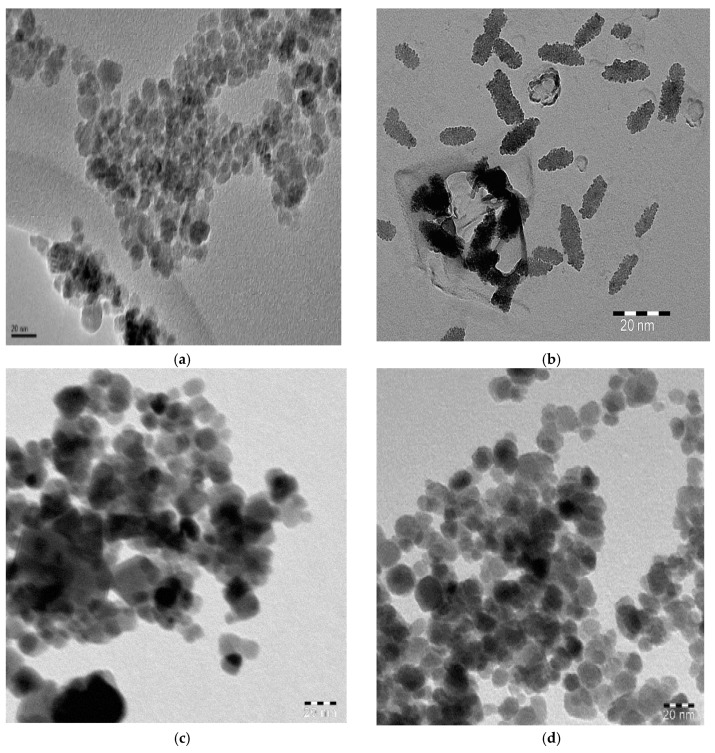
TEM micrographs of the prepared MNPs (**a**) absence of Myrrh, (**b**) presence of Myrrh, (**c**) presence of hydrophilic Myrrh and (**d**) presence of hydrophobic Myrrh extracts.

**Figure 4 nanomaterials-10-00262-f004:**
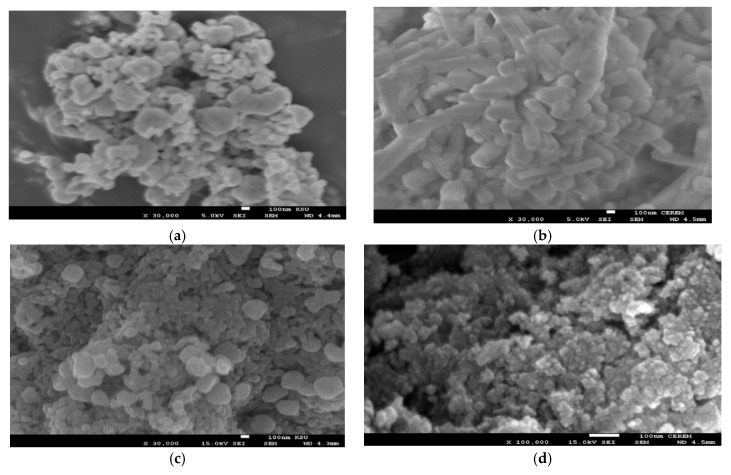
SEM micrographs of the prepared MNPs (**a**) absence of Myrrh, (**b**) presence of Myrrh, (**c**) presence of hydrophilic Myrrh and (**d**) presence of hydrophobic Myrrh extracts.

**Figure 5 nanomaterials-10-00262-f005:**
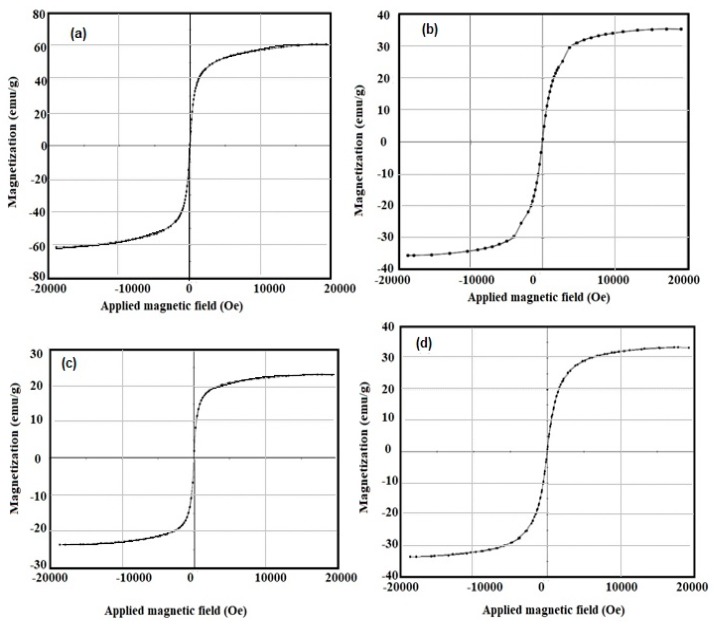
VSM hysteresis of MNPs (**a**) absence of Myrrh, (**b**) presence of Myrrh, (**c**) presence of hydrophilic Myrrh and (**d**) presence of hydrophobic Myrrh extracts.

**Figure 6 nanomaterials-10-00262-f006:**
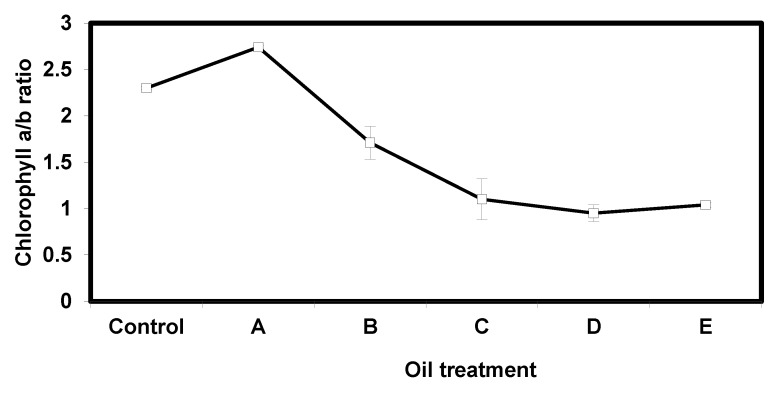
Concentration of carotenoid pigment in the leaf blades of *Eichhornia crassipes* plants in different levels of oil contamination compared to the control. A: (plant + 0.5 mL crude oil); B: (plant + 1 mL crude oil), C: (plant + 2 mL crude oil); D: (plant + 3 mL crude oil), E: (plant + 5 mL crude oil). Data are represented as mean ± SD *(n = 3).*

**Figure 7 nanomaterials-10-00262-f007:**
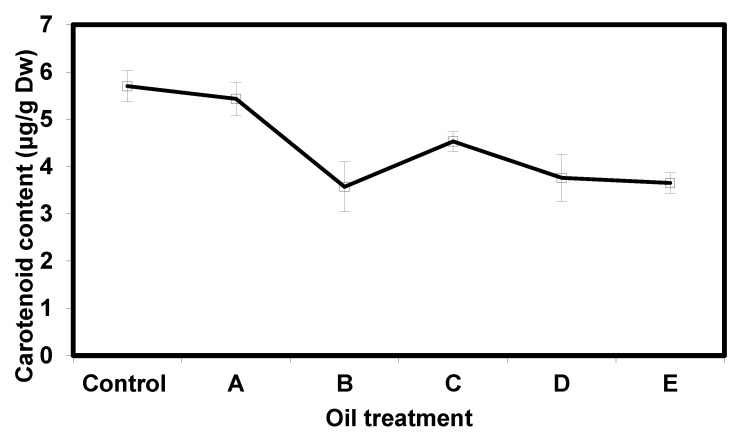
Concentrations of chlorophyll a and chlorophyll b in *Eichhornia crassipes* plants at different levels of oil contamination compared to the control. A: (plant + 0.5 mL crude oil); B: (plant + 1 mL crude oil), C: (plant + 2 mL crude oil); D: (plant + 3 mL crude oil), E: (plant + 5 mL crude oil). Data are represented as mean ± SD *(n = 3).*

**Figure 8 nanomaterials-10-00262-f008:**
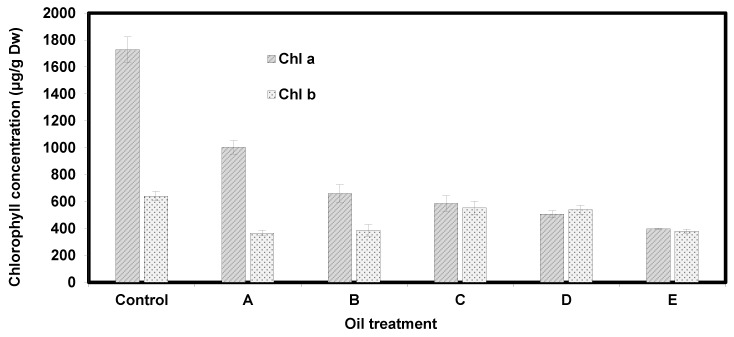
Ratio of chlorophyll a to chlorophyll b in *Eichhornia crassipes* plants compared to the control A: (plant + 0.5 mL crude oil); B: (plant + 1 mL crude oil), C: (plant + 2 mL crude oil); D: (plant + 3 mL crude oil), E: (plant + 5 mL crude oil). Data are represented as mean ± SD *(n = 3)*.

**Figure 9 nanomaterials-10-00262-f009:**
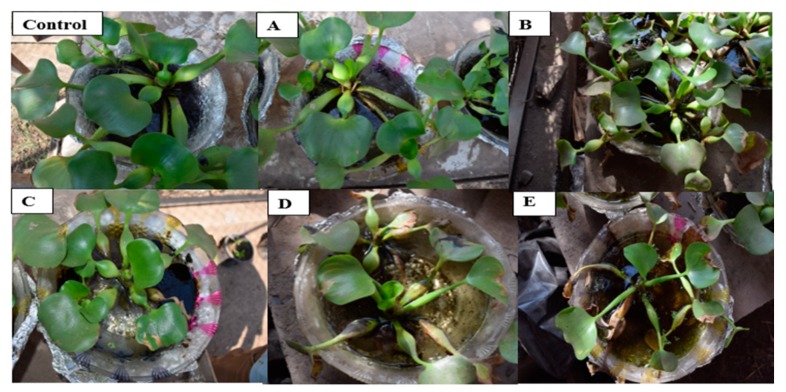
Morphological assessment of *Eichhornia crassipes*
**A**: (plant + 0.5 mL crude oil); **B**: (plant +1 mL crude oil), **C**: (plant + 2 mL crude oil); **D**: (plant + 3 mL crude oil), **E**: (plant + 5 mL crude oil).

**Figure 10 nanomaterials-10-00262-f010:**
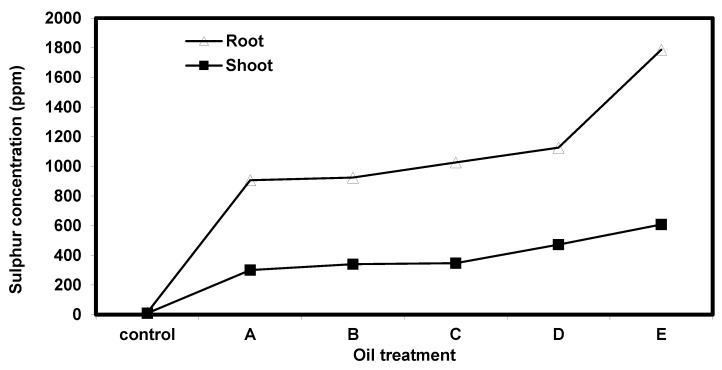
Sulphur content (ppm)in the root and shoot of measured in the root and shoot of *Eichhornia crasspises* plants grown for one month in water contaminated with varying levels of crude oil. A: (plant + 0.5 mL crude oil); B: (plant +1 mL crude oil), C: (plant + 2 mL crude oil); D: (plant + 3 mL crude oil), E: (plant + 5 mL crude oil).

**Figure 11 nanomaterials-10-00262-f011:**
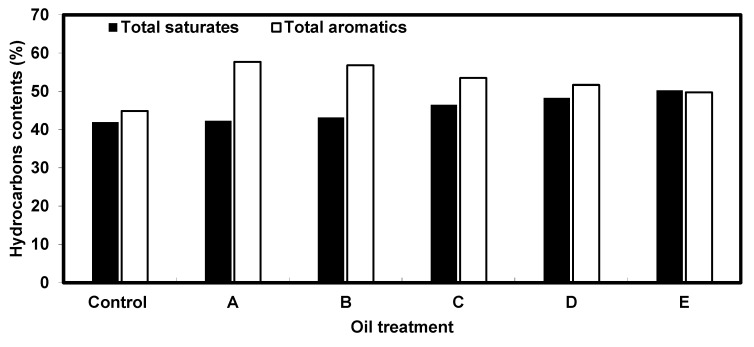
Concentrations of saturated and aromatic hydrocarbons measured in *E.crassipes* remediated water compared to baseline values. Baseline values: amount of saturated and aromatic hydrocarbons in oil contaminated water without the plant. **A**: (plant + 0.5 mL crude oil); **B**: (plant +1 mL crude oil), **C**: (plant + 2 mL crude oil); **D**: (plant + 3 mL crude oil), **E**: (plant + 5 mL crude oil).

**Table 1 nanomaterials-10-00262-t001:** Physiochemical properties of Ras Tanura Saudi heavy crude oil.

Test	Method	Result	
		Arabic Medium Crude	Egyptian Heavy Crude Oil
API gravity	Calculated	27.4	19
Specific gravity60/60(1F)	ASTMD-1298	0.893	0.934
Wax content (wt%)	UOP 46/64	5	2.3
Asphaltene content (wt%)	IP 143/84	13	18.3
Pour point(°C)	IP 15/67(86)	18	6

**Table 2 nanomaterials-10-00262-t002:** Magnetic properties for MNPs prepared in the absence and presence of Myrrh extracts at 25 °C.

MNPs	Magnetic Properties		
	Ms (emu/g)	Mr (emu/g)	Hc (G)
Iron oxide	60.68 ± 0.5	0.56 ± 0.03	14.50 ± 0.3
Myrrh	35.30 ± 0.3	3.80 ± 0.04	20.71 ± 0.2
Hydrophilic Myrrh	23.3 ± 0.6	5.70 ± 0.09	13.13 ± 0.4
Hydrophobic Myrrh	33.41 ± 0.7	0.19 ± 0.01	9.04 ± 0.1

**Table 3 nanomaterials-10-00262-t003:** Oil spill collection results of Arabic heavy crude oil.

MNPs	MNPs: Crude Oil Ratios (Wt.:Wt.)			
	1:1	1:10	1:25	1:50
Hydrophobic Myrrh extract	95 ± 1	92 ± 2	85 ± 3	80 ± 1
Myrrh	80 ± 3	70 ± 4	60 ± 5	45 ± 4

**Table 4 nanomaterials-10-00262-t004:** Summary for comparison of crude oil and remained residual samples on the basis detection of m/z 113 and 183.

Parameter	Constituents	Mass (m/z)	
		113	183
∑n-alkane/∑iso-alkane	Crude oil	2.63 ± 0.03	0.93 ± 0.01
	Remained residual	0.41 ± 0.01	0.11 ± 0.009
Pristane/phytane	Crude oil	1.09 ± 0.01	1.93 ± 0.02
	Remained residual	1.09 ± 0.02	1.91 ± 0.01
∑odd n-alkane/∑ even iso-alkane	Crude oil	0.93 ± 0.01	1.25 ± 0.001
	Remained residual	0.13 ± 0.01	0.29 ± 0.003

**Table 5 nanomaterials-10-00262-t005:** Crude oil polyaromatic, diaromatic and monoaromatic analysis.

	Polyaromatics	Diaromatics	Monoaromatics
Baseline Values	44.86	8.13	5.06
A	39.9	11.5	6.3
B	30.2	15.4	11.2
C	26.5	11.8	15.2
D	18.3	9.2	24.2
E	19.53	10	20.22

Baseline values: amount of saturated and aromatic hydrocarbons in oil contaminated water without the plant. A: (plant + 0.5 mL crude oil); B:(plant + 1 mL crude oil), C: (plant + 2 mL crude oil); D: (plant + 3 mL crude oil), E: (plant + 5 mL crude oil).

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
