# Peer review of "Green Technology for Remediation of Water Polluted with Petroleum Crude Oil: Using of Eichhornia crassipes (Mart.) Solms Combined with Magnetic Nanoparticles Capped with Myrrh Resources of Saudi Arabia"

_nanomaterials, 2020, doi:10.3390/nano10020262_

Round 1

Author Response

Reviewer 1

line 27: "Magnetic nanoparticles were used to extract the heavy components of the crude oils. " - it is unclear - at what point nanoparticles were used - it is the abstract and it should be informational for wide audience

Answer: Magnetic nanoparticles based on magnetite capped with Myrrh extracts were prepared, characterized and used to collect the heavy components of the crude oils.

line 48: space between 2017 and contamination

answer: It is corrected

First of all, the Authors should say what is the scientific novelty? What was the research hypothesis? This information should appears also in the abstract.

Answer: They inserted at the end of introduction and abstract.

Advantages of using magnetic nanomaterials should be presented. We introduce additional factors into the environment and their impact on the environment should be presented.

Answer: Their importance were clarified.

n-hexane, n-heptane according IUPAC nomenclature should be writtten hexane, heptane

Answer: All modified to be hexane and heptane

line 112, 126: please remove the space between the value and the percentage.

Answer: they were corrected

line 149-151, 593: please introduce the space between the value and ml.

answer: the space inserted

Whether the water used to maintain the plants was analyzed?

Answer: the salinity of seawater analyzed and the remained hydrocarbon contaminates were analyzed in water while only sulfur analyzed in the plant

line 216: subscript in “FeCl3”

answer: it is corrected

FTIR spectra -figure 1 b,c, d - the wide peak at ca. 3400 suggest present of intermolecular hydrogen bonds - was water present in this samples?

Answer: the samples were dried and capped with hydrophobic extract that cannot disperse in water. This peak was referred to hydroxyl groups on the magnetite surfaces.

Figure 11 - please add error bars

answer: all figures modified

Generally the quality of the figures should be improved.

Answer: the quality and resolution were improved.

Please consistent the description of the tables in the same way.

Answer: the tables discussed in the text.

Cyclo-paraffin and isoparaffin should be written together

answer: cycloparaffins are different thane n-paraffin and iso-paraffin which are main constituent of the crude oil.

line e.g. 582, 598 “E.crassipes” - should be space.

Answer: it is modified

How is the practical application of this work

answer: new paragraph and reference added

Reference formatting needs improvement. Please provide correct transcript of name of the journal, eg. is Journal of hazardous materials should be Journal of Hazardous Materials

answer: the references formatted according nanomaterial format.

Reviewer 2 Report

1) The abstract contains a lot of general information. Please start with ''The main aim of this work …...'' and write only the information about the work done for this paper.

2) What are the factors that cause Crude oil contamination?

3) What is the typical composition of the contaminants present and its sources in crude oil. Provide examples and references. 

4) The role of degree of salinity and water temperature should be discussed in more detail, as it affects the biological activity and also the species of chemicals.

5) Introduce the efficiency of different  oil-spill collectors used in this process of decontamination. Provide case studies where they have been used to solve real problems.

6) What are the limitations of using phyto-techniques for oil spill clean up. What are the success stories in practice? Give examples and references from scientific literatures.

7) What are the threshold values for growth rate, large biomass, and tolerance of water hyacinth towards many metal and metalloid elements. Give examples.

8) What are the physico-chemical properties of Myrrh gum.

9) Write the specific objectives of this study clearly.

10) Provide details of all the instruments and analytical equipments used, its model number, country of manufacture.

11) How was QA ensured when collecting samples from the Nile river?

12) Many sentences are INCOMPLETE in this manuscript. The authors have not checked the document carefully. More than 25 such mistakes can be found:

Example: Arabic heavy and medium crude produced from ARAMCO (ARAMCO; Kingdom of Saudi Arabia)

13) The quality of Figures 1 and 2 is NOT GOOD at all. Please enhance its quality OR remove them from the manuscript. All the text in the figure should be readable when printed in Black and White A4 paper.

14) Write and discuss the role of hydrophilic and hydrophobic extracts of Myrrh on the results of this study.

15) What is the role of carboxylic, hydroxyl, aldehyde and amine functional groups with the hydroxyl groups on the decontamination process. facilitate good scientific mechanism based discussion.

16) Between number and unit, there has to be a space. There are more than 50 such mistakes.

17) Remove the border in Figures 6, 7 and 8. Enhance its quality. They are VERY DULL at the moment.

18)  The spelling of chlorophyll is wrong in the figure. USE ARIAL font in the x and y axis of all the figures. Check the entire document for such mistakes.

19) Figures 10 and 11 quality is ALSO BAD.

20) Divide the discussions section with respect to the specific objectives of this manuscript and facilitate more discussion. Refer to more recent references in this section.

21) Write the practical application of this work and its limitations in 1 nice paragraph.

22) Reference formatting is completely inconsistent. A lot of mistakes can be seen. Even the journal names are NOT written properly. 

Author Response

Comments and Suggestions for Authors

The abstract contains a lot of general information. Please start with ''The main aim of this work …...'' and write only the information about the work done for this paper.

Answer: The general information deleted and the abstract modified.

What are the factors that cause Crude oil contamination?

Answer: introduction represented the environmental impact of the crude oil contaminate and the present work aims to use natural material by removal of crude oil contaminate by using magnetic nanomaterials capped with natural products followed by using the plant to absorb light hydrocarbons.This technology does not include the bioremidatiom of the hydrocarbons into small fragments by enzymes or bacteria. This work study only the effect of the crude oil concentrations on the uptake either by magnetic nanoparticles or by Using of Eichhornia crassipes (Mart.) Solms

What is the typical composition of the contaminants present and its sources in crude oil. Provide examples and references. 

Answer: Table 1 inserted to clarify the crude oil specs.

The role of degree of salinity and water temperatureshould be discussed in more detail, as it affects the biological activity and also the species of chemicals.

Answer: The present work aims to use the plat for absorb the crude oil without degradation. And the conditions illustrated in the experimental section 

Introduce the efficiency of different oil-spill collectors used in this process of decontamination. Provide case studies where they have been used to solve real problems.

Answer: This technology is new and it is applied on labe scale for the academic research.

What are the limitations of using phyto-techniques for oil spill clean up. What are the success stories in practice? Give examples and references from scientific literatures.

Answer: At the time of this writing, the consequences of the technologies applied to contain the oil and to prevent its reach into sensitive coastal and wetland habitats, as well as the response of the ecosystem to the spill, have not been, and may never be, fully documented. Work to understand the fate and effects of the spilled oil is ongoing. Much work to determine the long-term effects remains. Understanding of the potential impacts of response technologies to ecosystem services and of the impacts of oil on multiple layers of ecosystems and their services is limited by the availability of baseline information and of important data collected during the process. Until this massive and unprecedented collection of data is complete and the results made available, any assessment of the impacts of the response technologies will not be fully informed.

7) What are the threshold values for growth rate, large biomass, and tolerance of water hyacinth towards many metal and metalloid elements. Give examples.

8) What are the physico-chemical properties of Myrrh gum.

Answer: The chemical composition of Myrrh discussed in the experimental, results and discussion section.

9) Write the specific objectives of this study clearly.

Answer: The objectives clarified at the end of the introduction section.

10) Provide details of all the instruments and analytical equipments used, its model number, country of manufacture.

Answer: The details of the equipment and their model number were inserted.

11) How was QA ensured when collecting samples from the Nile river?

12) Many sentences are INCOMPLETE in this manuscript. The authors have not checked the document carefully. More than 25 such mistakes can be found:

Example: Arabic heavy and medium crude produced from ARAMCO (ARAMCO; Kingdom of Saudi Arabia)

Answer: The mistakes revised and corrected.

13) The quality of Figures 1 and 2 is NOT GOOD at all. Please enhance its quality OR remove them from the manuscript. All the text in the figure should be readable when printed in Black and White A4 paper.

Answer: The quality and resolution of All Figures were improved.

14) Write and discuss the role of hydrophilic and hydrophobic extracts of Myrrh on the results of this study.

Answer: New paragraph added to clarify the chemical structures and composition of Myrrh

15) What is the role of carboxylic, hydroxyl, aldehyde and amine functional groups with the hydroxyl groups on the decontamination process. facilitate good scientific mechanism based discussion.

Answer: The role of Myrrh chemical composition on the hydrophobicity of magnetite and morphology discussed to facilitate the dispersion of the particle into crude oil without precipitation in water to facilitate the collecting of the crude oil.

16) Between number and unit, there has to be a space. There are more than 50 such mistakes.

Answer: They all revised and corrected.

17) Remove the border in Figures 6, 7 and 8. Enhance its quality. They are VERY DULL at the moment.

Answer: The quality and resolution of All Figures were improved

18)  The spelling of chlorophyll is wrong in the figure. USE ARIAL font in the x and y axis of all the figures. Check the entire document for such mistakes.

Answer: The Figures revised improved.

19) Figures 10 and 11 quality is ALSO BAD.

Answer: The quality and resolution of All Figures were improved.

20) Divide the discussions section with respect to the specific objectives of this manuscript and facilitate more discussion. Refer to more recent references in this section.

Answer: The discussion divided into two separate sections to clarify the objective of the work and the references cited the recent works

21) Write the practical application of this work and its limitations in 1 nice paragraph.

Answer: New paragraphs added to clarify the importantance of the work as

22) Reference formatting is completely inconsistent. A lot of mistakes can be seen. Even the journal names are NOT written properly. 

Answer: The references revised according to nanomaterial format

Reviewer 3 Report

I think it's a very interesting article.

Author Response

The article revised and the new changes marked with the red colour to improve the quality

Round 2

Reviewer 1 Report

The Authors introduced appropriate changes, however some minor points should be improved:
1. line 28 - please delete the space between 95 and %.
2. Tables title should be write in the same way: bold or not etc.
3. e.g. line 356-357, 366 - please introduce the space between the value and ml.
4. In the manuscript please write down consonantly ml or mL.
5. line 596: Gypsophilastruthium, Helianthemunsquamatum?

Author Response

Comments and Suggestions for Authors

line 28 - please delete the space between 95 and %.

Answer: the spaces removed.

Tables title should be write in the same way: bold or not etc.

Answer: Table format unified.

e.g. line 356-357, 366 - please introduce the space between the value and ml.

Answer: All space modified.

In the manuscript please write down consonantly ml or mL.

Answer all mL.

line 596: Gypsophilastruthium, Helianthemunsquamatum?

Answer: The space deleted to be Gypsophila struthium, Helianthemun squamatum

Reviewer 2 Report

1) Its indeed very sad to see that the authors were merely in a HURRY to answer the comments of the reviewer, without even caring for the grammar mistakes.

Example: The present work aims to use the plat for absorb the crude oil without degradation

Check the spelling of lab .... in the authors response file. It has been written as ''labe''.

Similarly, there are more than 30-35 mistakes with formatting and also spacing, English language mistakes. 

2) Write details about the quality assurance of the samples and its statistical analysis.

3) A lot of sentences in this document has sentence structure errors:

I am just showing two examples:

......... Saudi Arabia improved for remediation of highly polluted water by crude petroleum oil.

........ Water hyacinth used to degrade polycyclic aromatic hydrocarbons from soil leachate and achieved acceptable results.

Similarly, each and every sentence has to be proofread by someone who is familiar with the topic and also someone who can check word by word.

4) What is the implication of acute toxicity for this study? Explain its role clearly from a legislative and health viewpoint.

5) How does the specific gravity values affect the mobility of the contaminant in water bodies?

6) Format the equation numbers properly and refer to the equations in the text.

7) Write the statistical terms such as F, P etc in italics.

8) The role of polysaccharides, amino acids and proteins should be explained clearly.

9) What was the stability of the Fe-O?

10) Figure 1 - The numbers in the X and Y axis of the graphs are VERY SMALL. I had printed this document and I cannot read it properly. Can the authors print and check this?

11)  The hydrophobicity of MNPs should be discussed further and how did contribute to the remediation of the petroleum crude oil.

12) provide all the reactions involved in the degradation of crude oil.

13) What are the end products of crude oil degradation? Where can we find that information?

14) What is the proof to say that this technology is GREEN and what are the general criteria?

15) Figure 5 is stretched and looks VERY BAD.

16) The entire results and discussions has a lot of text that belongs to the materials and methods:

Example: The oil spill was collected by using an external magnet and
washed with ethanol followed by chloroform and reused five times without change in the CE%. 

Kind note: Its sad to see that the authors did not even write the section contents properly. All the words have to be re-read and checked again. In the results and discussion, ONLY write the results and discussion.

17) Between the number and unit, there is space. There are more than 50 mistakes.

18) Figure caption is BELOW and not above. Check all the mistakes.

19) The entire document has more than 100 spacing mistakes. I am showing only one:

Example: n chlorophyll b.     Eichhornia crassipes plants 

20) What are the factors that affect the S content in the root and shoot of Eichhornia crassipes?

21) In Y ais, write the unit within brackets. Check all the figures and be consistent.

22) How are cyclo-paraffin and iso-paraffin removed from the crude oil ?

23) What is the meaning of operational end point?

24) In the literature, are there reports related to sensitive coastal habitats and how contamination has been removed? Provide examples.

25) Give examples for phytoaccumulation of sulphur with references.

26) References has more than 25 mistakes. Nothing seems to be working properly for this manuscript.

Example: Cyperus brevifolius is not written in italics.

Check word by word and format the entire references MANUALLY.

Kind note:

a) If the next version of the document contains any formatting, spacing, and English mistakes - it can be rejected.

b) The authors are advised to implement all the above-mentioned comments in the revised manuscript and the new changes made should be highlighted in BLUE coloured text so that the differences can be seen clearly in Revision 2.

Author Response

Reviewer 2

1) Its indeed very sad to see that the authors were merely in a HURRY to answer the comments of the reviewer, without even caring for the grammar mistakes.

Example: The present work aims to use the plat for absorb the crude oil without degradation

Check the spelling of lab .... in the authors response file. It has been written as ''labe''.

Answer: We apologize for the mistakes send to the reviewer report that did not included in the article document.

Similarly, there are more than 30-35 mistakes with formatting and also spacing, English language mistakes. 

Answer: The article reviewed and the corrections marked with the blue color.

2) Write details about the quality assurance of the samples and its statistical analysis.

Answer: Statistical analysis was performed using one-way Analysis of Variance (ANOVA) and Least Significant Difference (LSD) test to determine the differences between group’s mean and standard error at 0.05 levels. Frequencies were computed and Chi-square test was utilized to study the association between the different variables. All statistics were carried out using Statistical Analysis Systems (SAS) program.

3) A lot of sentences in this document has sentence structure errors:

I am just showing two examples:

......... Saudi Arabia improved for remediation of highly polluted water by crude petroleum oil.

Answer: The sentence modified as

The presence of magnetite nanoparticles capped with Myrrh resources improved the remediation of highly polluted water with the petroleum crude oil

........ Water hyacinth used to degrade polycyclic aromatic hydrocarbons from soil leachate and achieved acceptable results.

Similarly, each and every sentence has to be proofread by someone who is familiar with the topic and also someone who can check word by word.

Answer: The article reviewed and the corrections marked with the blue color.

4) What is the implication of acute toxicity for this study? Explain its role clearly from a legislative and health viewpoint.

Answer: The acute toxicity of the crude oil for marine environment well known. The present work related to apply plant to absorb crude . The following text included references added as

Crude oil is well known for its phytotoxicity and capacity to alter morphological, physiological, and biochemical properties of plants. It was previously reported that, microcosm experiments performed in situ in North Inlet Estuary near Georgetown, South Carolina, using Texas crude oil confirmed a decrease in chlorophyll a in phytoplankton as crude oil concentration increased from 10 to 100 microliters per liter [22].

5) How does the specific gravity values affect the mobility of the contaminant in water bodies?

The mobility of the heavy and petroleum crude oils having higher specific gravity improved in the presence of nanomaterials and leads to an improvement in oil viscosity and recovery [46]. Accordingly, the remained residual of the petroleum crude oil contaminates can be easily phytoremediated by using water hyacinth plants.

6) Format the equation numbers properly and refer to the equations in the text.

Answer: The equation number inserted in the text.

7) Write the statistical terms such as F, P etc in italics.

Answer: They modified.

8) The role of polysaccharides, amino acids and proteins should be explained clearly.

Answer:

It was suggested that the presence of carbonyl groups (ester, aldehyde and ketone) in the chemical structures of the capping agent improved their binding with magnetite nanoparticles due to their electron-withdrawing effect. This binding increases the water dispersibility of magnetite NPs [8]. Moreover, the primary alcohol groups of polysaccharides were partially oxidized to the resultant carbonyl groups that provide sufficient protection and stability to magnetite nanoparticles beside their high water-dispersibility [7].

9) What was the stability of the Fe-O?

Answer: The disappearance of Fe-O peaks of other iron oxides (750-950 cm-1) elucidates that the Myrrh extract protected the magnetite to oxidize further for another iron oxide such as maghemite and hematite [6]. The reusability of the prepared MNPs for several cycles elucidate their high stability to salinity of seawater and crude oil acidity

10) Figure 1 - The numbers in the X and Y axis of the graphs are VERY SMALL. I had printed this document and I cannot read it properly. Can the authors print and check this?

Answer: Figure 1 modified and clarified

11)  The hydrophobicity of MNPs should be discussed further and how did contribute to the remediation of the petroleum crude oil.

Answer: The Hydrophobicity of MNPs assists only their good dispersion in the crude oil only and facilitate the collection of oil-spill and reduce the remained crude oil amount in the polluted water.

12) provide all the reactions involved in the degradation of crude oil.

Answer: The present work used the plant to absorb the crude oil and it was not used for the crude oil degradation.

13) What are the end products of crude oil degradation? Where can we find that information?

Answer: the crude oil not degraded in the present work but collected by MNPs and absorbed by Eichhornia crassipes (water hyacinth).

14) What is the proof to say that this technology is GREEN and what are the general criteria?

Answer: The green technology of the present work was referred to the using and preparing of ecofriendly chemicals based on cheap natural resources, and nontoxic materials based on magnetite, Myrrh and Eichhornia crassipes (water hyacinth) to remove hazardous and toxic petroleum crude oil water pollutants.

15) Figure 5 is stretched and looks VERY BAD.

Answer: Figure 5 clarified

16) The entire results and discussions has a lot of text that belongs to the materials and methods:

Example: The oil spill was collected by using an external magnet and
washed with ethanol followed by chloroform and reused five times without change in the CE%. 

Answer: The sentence corrected to be

The oil spill was reused, as described in the experimental section, five times without change in the CE%.

Kind note:  All the words have to be re-read and checked again. In the results and discussion, ONLY write the results and discussion.

Answer: The article revised, corrected and marked with the blue color.

17) Between the number and unit, there is space. There are more than 50 mistakes.

Answer: All mistakes were corrected in the first version.

18) Figure caption is BELOW and not above. Check all the mistakes.

Answer: All Captions are inserted below.

19) The entire document has more than 100 spacing mistakes. I am showing only one:

Example: n chlorophyll b.     Eichhornia crassipes plants

Answer: The space deleted to be Gypsophila struthium, Helianthemun squamatum and all spaces corrected and modified.

20) What are the factors that affect the S content in the root and shoot of Eichhornia crassipes?

Answer: Absorption of crude oil degradation materials such as sulphur and other pollutants from the water medium depends mainly on plant roots which absorb and accumulate the pollutants and subsequently translocate it into other plant organs.

21) In Y ais, write the unit within brackets. Check all the figures and be consistent.

Answer: All Figures checked and corrected.

22) How are cyclo-paraffin and iso-paraffin removed from the crude oil?

Answer: New reference 15 added to confirm that the Field ionization mass spectrometry (FIMS) is a recommended technique to analyze complex multicomponent hydrocarbon mixtures (n-paraffins, iso-paraffins and cycloparaffins) without using preior chromatographic separation.

23) What is the meaning of operational end point?

Answer: Operational end point mean the positive and negative factors affected the remediation operation management include all analytics, remote collectors, and load balancer.

24) In the literature, are there reports related to sensitive coastal habitats and how contamination has been removed? Provide examples.

Answer: Contamination of these water resources by petroleum hydrocarbons leads to significant consequences on human health and on biotic components of the ecosystem and is thus considered a global problem that requires intervention. The contamination of surface and groundwater is worldwide problem that affects biodiversity, undermines economic growth and the health of billions of people [2]. The impact of oil on marine environments is influenced by several factors, including the amount of oil, its chemical composition, the abundance of the toxic compounds, and the proximity to environmentally, economically sensitive areas [1]. There are several strategies such as physical, mechanical and chemical treatments, which are used to alleviate the pollution of the crude oil contaminants; though, the aforementioned treatments present consequences that are harsh on the environment and occasionally lack efficacy in action. Physical or mechanical removal provides only partial removal, whereas chemical-based methods result in additional toxic effects on the environment. Environmental research developments, particularly green technology proved to be an environmentally safe alternative for the remediation of polluted areas.

25) Give examples for phytoaccumulation of sulphur with references.

Answer: Absorption of the petroleum crude oil materials such as organic sulphur and other pollutants from the water medium depends mainly on plant roots, which absorb and accumulate the pollutants and subsequently translocate it into other plant organs. Microorganisms associated with plant play a role in oil degradation and positively affect the absorption and accumulation of sulphur in plant roots [37].

26) References has more than 25 mistakes. Nothing seems to be working properly for this manuscript.

Example: Cyperus brevifolius is not written in italics.

Check word by word and format the entire references MANUALLY.

Answer: All references modified and marked with blue color.

Round 3

Reviewer 2 Report

1) Remove the word ''BY'' before the authors names.

2) Provide postal code for the addresses.

3) Has water hyacinth been applied for any other environmental remediation purposes in Egypt? Provide examples from your home country.

4) Write hours as just h, seconds as s, minutes as min.

5) Make sure that the alignment of placing a, b, c, d of the figures is in a straight line.

6) Numbering the equations also should be in a straight line.

7) Remove the horizontal lines in the tables. Check how tables are drawn professionally.

8) Figure 5 - make them all in uniform size. One of them is small.

9) Even in this version, some sentences has no clar meaning.

Example: Plants subjected to 1 mL, 2 mL, 3 mL and 5 mL of crude oil all showed ......

Check all the document one more time for English errors.

10) Figure 8 in the PDF is so bad. Why?

11) Entire figures should be checked in the PDF generated. Its very bad and overlapping with the text. I am not sure why did the authors approve this PDF and submit. Its very sad. Indeed, the carelessness of the authors and I am not sure why the authors are in such a HURRY.

12) What did references [3-10] say about  hydrophobic capping agents? Please write 5-6 more lines of text in the discussion. This should be added to explain the mechanisms.

13) Magnetite and Myrrh extract are non-toxic? Show evidence based on its chemical characteristics.

14) According to [3-] what was the reason for progressive reduction in chlorophyll b content ? Explain more clearly.

15) Provide reasons for all the discussions and dont merely mention which authors reported what! Describe it more clearly. The readers cannot go and read all the references.

Kind note:

References needs to be checked word by word. There are more than 55 mistakes. I am so sorry. This is very sad.

Example: (Solanum lycopersicum) not in italics.

Why are the authors using unwanted upper case when writing the titles of articles? Check word by word. 

I am of the impression that the authors JUST DONT CARE for formatting and please understand that the reviewers and editors of MDPI are not here to correct the author's careless mistakes. I am now wondering, if, all the authors really contributed to this manuscript. The manuscript shall be rejected if there are such formatting mistakes in the next VERSION.